# Radiotherapy Combined with PD-1 Inhibition Increases NK Cell Cytotoxicity towards Nasopharyngeal Carcinoma Cells

**DOI:** 10.3390/cells10092458

**Published:** 2021-09-17

**Authors:** Anna Makowska, Nora Lelabi, Christina Nothbaum, Lian Shen, Pierre Busson, Tram Thi Bao Tran, Michael Eble, Udo Kontny

**Affiliations:** 1Division of Pediatric Hematology, Oncology and Stem Cell Transplantation, Medical Faculty, Rhenish-Westphalian Technical University, 52074 Aachen, Germany; amakowska@ukaachen.de (A.M.); nora.lelabi@rwth-aachen.de (N.L.); cnothbaum@ukaachen.de (C.N.); lshen@ukaachen.de (L.S.); 2CNRS UMR 8126, Gustave Roussy, Université Paris Sud, Université Paris-Saclay, 94805 Villejuif, France; Pierre.BUSSON@gustaveroussy.fr (P.B.); TH.TRAN@gustaveroussy.fr (T.T.B.T.); 3Department of Radiation Oncology, Medical Faculty, Rhenish-Westphalian Technical University, 52074 Aachen, Germany; meble@ukaachen.de

**Keywords:** nasopharyngeal carcinoma, natural killer cells, programmed cell death ligand 1, radiotherapy, nuclear factor kappa B, interferon-beta

## Abstract

Background: Nasopharyngeal carcinoma (NPC) in endemic regions and younger patients is characterized by a prominent lymphomononuclear infiltration. Radiation is the principal therapeutic modality for patients with NPC. Recent data suggest that the efficacy of radiotherapy in various cancers can be augmented when combined with immune checkpoint blockade. Here, we investigate the effect of radiotherapy on the killing of NPC cells by Natural Killer (NK) cells. Methods: NPC cell lines and a patient-derived xenograft were exposed to NK cells in the context of radiotherapy. Cytotoxicity was measured using the calcein-release assay. The contribution of the PD-L1/PD-1 checkpoint and signaling pathways to killing were analyzed using specific inhibitors. Results: Radiotherapy sensitized NPC cells to NK cell killing and upregulated expression of PD-1 ligand (PD-L1) in NPC cells and PD-1 receptor (PD-1) in NK cells. Blocking of the PD-L1/PD-1 checkpoint further increased the killing of NPC cells by NK cells in the context of radiotherapy. Conclusion: Radiation boosts the killing of NPC cells by NK cells. Killing can be further augmented by blockade of the PD-L1/PD-1 checkpoint. The combination of radiotherapy with PD-L1/PD-1 checkpoint blockade could therefore increase the efficacy of radiotherapy in NPC tumors.

## 1. Introduction

Nasopharyngeal carcinoma (NPC) is a malignancy arising from the epithelium of the nasopharynx. The major etiological factor proposed for NPC pathogenesis in patients in endemic regions and in younger patients worldwide is Epstein-Barr virus (EBV) infection [1,2,3]. Most NPCs are radiosensitive. Because of its anatomic characteristics, NPC is rarely treated surgically and radiotherapy with doses between 66 and 70 Gy is the major therapeutic modality for adult patients with NPC [4,5]. The introduction of chemotherapy into NPC treatment protocols for children and adolescents has allowed the reduction of the radiation dose in this age group to 60 Gy and lower [6,7,8]. Unfortunately, radiotherapy can produce undesirable complications due to the proximity of the tumor to critical structures. About 70% of patients suffer from major late effects such as xerostomia, endocrinopathies, or secondary neoplasms [9,10,11]. Since the frequency and intensity of side effects correlate with the radiation dose, the development of a radiation sensitizer to further lower the radiation dose could be a useful means to decrease side effects while preserving tumor control.

NPC tumors are characterized by an immune cell infiltrate, including NK cells. Several studies have demonstrated that increased NK cell infiltration closely relates to a favorable prognosis [12,13,14]. In our previous studies, we showed that NPC cells are sensitive to killing by NK cells and that activation of NK cells by interferon beta (IFNβ) increased their cytolytic capacity against NPC cells primarily by induction of the death ligand TRAIL [15,16]. 

Recently, it has been observed that the efficacy of radiotherapy in various cancers can be augmented when combined with immune checkpoint blockade [17]. Notwithstanding that the response of NPC to radiotherapy has been well characterized, it is unclear how radiotherapy may impact immune-mediated recognition of cancer. Furthermore, little is known regarding the effect of radiotherapy on NK cell recognition or cytolysis of nasopharyngeal carcinoma cells. 

Here, we have investigated the effect of radiotherapy on the killing of NPC cells by NK cells.

## 2. Materials and Methods

### 2.1. Cell Lines

Four NPC cell lines C666-1, C17, HK1 and TW01 were used. Cell line C666-1 was a gift from Prof. Fei-Fei Liu, University of Toronto, Canada [17]. Cell line TW01 was supplied by Prof. Chin-Tarng Lin (National Taiwan University Hospital) [18]. Cell line C17 was a gift from Prof. Sai Wah Tsao (Chinese University of Hong Kong, China) and had been established from a C17 xenograft [19]. Cell line HK1 was obtained from Prof. Kwok Wai Lo from the Chinese University of Hong Kong, China [20]. Cell authentication was done using short tandem repeated profiles as described previously [21], and cell lines were tested at regular intervals by PCR to rule out mycoplasma contamination. The C17 cell line was authenticated by checking HLA class I alleles on PCR (A02.01/A26.01–B44.02/B51.01). Cell lines were grown in RPMI1640 medium (Gibco; Paisley, UK) supplemented with 10% fetal bovine serum (Gibco; Paisley, UK) and 100 U/mL penicillin and 100 mg/mL streptomycin (Gibco; New York, NY, USA). Cells were cultured in a humidified incubator with 95% air and 5% CO_2_ at 37 °C.

### 2.2. Patient-Derived Xenograft

The C17 xenograft was established from a patient with an EBV-positive metastatic NPC. For the experiments described below, single-cell suspensions were derived from freshly isolated C17 tumor fragments by collagenase cell dispersion and cultured as described before [22].

### 2.3. Animal Studies

Swiss nude mice were bred in the animal facility at Gustave Roussy and housed in pathogen-free conditions in filter cap cages holding a maximum of five animals with irradiated aspen chip bedding and cotton fiber nesting material. They were maintained on a 12/12 light/dark cycle, with ad libitum UV-treated water and RM1 rodent diet. Typically, xenografts were performed on 6–8 female mice by the subcutaneous introduction of tumor fragments (about 200 mg) under general anesthesia. They were sacrificed when the total tumor volume reached 1700 mm^3^. The animals were monitored for signs of pain, such as immobility or restlessness, reduction of drinking, and food intake. The persistence of ab-normal behaviors led to the euthanasia of animals presumed to be suffering. Prior to tumor collection, mice were sacrificed by cervical dislocation. Otherwise, mice were euthanatized by carbon dioxide asphyxiation.

### 2.4. Isolation of Primary Human NK Cells

Human peripheral blood mononuclear cells (PBMC) were purified from buffy coats of four healthy donors using Ficoll-Hypaque (Biochrom; Berlin, Germany) density gradient centrifugation. Informed consent was obtained from all donors. NK cells were isolated from PBMC using a positive magnetic selection of CD56+ cells (Miltenyi; Bergisch Gladbach, Germany) as described before [15,16]. The purity of NK cells was >95% as determined by flow cytometric analysis of cells stained with anti-CD56-APC and anti-CD3-PerCP. CD3 contamination in purified NK cells was <1%.

Purified NK cells were cultivated in RPMI1640 medium supplemented with 10% FCS and 100 U/mL penicillin and 100 mg/mL streptomycin (Gibco) over night. NK cells were stimulated with recombinant human IFNβ (R&D System, New York, NY, USA) or irradiated as further indicated in the manuscript.

### 2.5. Reagents

Methylthiazolyldiphenyl-tetrazolium bromide (MTT) was obtained from Sigma Aldrich (St. Louis, MO, USA), human recombinant interferon beta from R&D System (New York; NY, USA), the PD-1 inhibitor nivolumab from Bristol-Myers (Anagni, Italy). Calcein-AM was purchased from Thermo Fisher (Eugene, OR, USA). Primary mouse monoclonal antibody against PD-1, clone 913429 was obtained from R&D System (Wiesbaden, Germany), anti-human B7-H1/PD-L1 monoclonal antibody, clone 130021 from R&D System (Minneapolis, MN, USA), anti-human B7-DC/PD-L2 Clone 176611 from R&D (Minneapolis, MN, USA) and mouse anti-human-RELA/NF-κB p65 (Clone 532301; R&D System). The APC-goat anti-mouse IgG secondary antibody was purchased from Santa Cruz Biotechnology (Heidelberg, Germany) and the PE-goat anti-mouse IgG secondary antibody was purchased from Biolegend (San Diego, CA, USA). All primary antibodies were diluted 1:100 in PBS. The APC-goat anti-mouse IgG secondary antibody and the PE-goat anti-mouse IgG secondary antibody were diluted 1:200 in PBS. Incubation time for all antibodies was at least 30 min. NF-κB inhibitor—BMS-345541 was obtained from Sigma Aldrich.

### 2.6. Cell Viability Assay

The MTT assay was used to determine the effect of radiation (0 to 6 Gy) on cell viability. The protocol was adopted from the literature [15,16]. Briefly, cells were seeded into 96-well plates at a density of 2500 cells/well within 200 µL of growth medium. After 24 h of culture, cells were irradiated and further incubated for 24, 48 and 72 h. At the end of the incubation periods, MTT solution was added. After 4 h, supernatant was removed and 200 μL of DMSO:isopropanol (1:1) solution was added to dissolve the formazan crystals. Thereafter, optical density was measured.

### 2.7. Cell Cycle Analysis

Propidium-iodide staining of nuclei was used to determine the effect of radiotherapy on cell cycle distribution as well as apoptosis by measurement of sub-G1 DNA content. After radiation (0 to 6 Gy), NPC cells were incubated up to 72 h. Cells were then processed as described previously [15,16]. Three independent experiments were performed for each assay.

### 2.8. Determination of Radiation Dose

Cells were exposed to photon-IR using a Faxitron X-ray irradiator (Munich, Germany), irradiated at a voltage of 160 kV and a current of 6.3 mA. The dose rate was 0.83 Gy/min. In order to identify the radiotherapy doses for the main experiments, NPC cells were irradiated from 0 to 6 Gy, and cell viability and apoptosis was measured. Two Gy was identified as dose significantly decreasing cell survival and inducing apoptosis in NPC cell lines and was thus chosen for further experiments (Appendix A).

### 2.9. Flow Cytometry

The surface expression of PD-L1, PD-L2 and PD-1 on NPC and NK cells was measured by flow cytometry as described before [15,16]. For detection of NF-κB expression, the cells were fixed and permeabilized with Cytofix/Cytoperm™ Fixation and Permeabilization Solution (BD Pharmingen; San Diego, CA, USA), then labeled with mouse anti-human NF-κB-antibody and afterward with APC-conjugated goat-anti-mouse antibody. Data were analyzed by the FlowJo software (FlowJo). Three independent experiments were performed for each assay.

### 2.10. Calcein Release Assay

NPC cells were used as targets and stained with fluorescence-based Calcein-AM for determining the cytotoxicity of NK cells. Target cells were resuspended at 1 × 10^6^ cells/mL in medium containing 15 µM Calcein-AM and incubated for 30 min at 37 °C. An NK:NPC ratio of 6:1 was chosen for the experiments in this paper, as we had previously demonstrated that NK cells induced high levels of calcein release in an E:T ratio-dependent manner in both NPC cell lines and PDX cells C17 [15]. The targets cells were then co-incubated with NK cells at an effector to target (E:T) ratio of 6:1 for 4 h at 37 °C or used as controls; 4% Triton (Merck; Darmstadt, Germany) was added to ensure maximum calcein release in controls while spontaneous calcein release was measured in NPC cells not co-incubated with NK cells. After co-incubation, relative fluorescence units (RFU) were measured from cell-free supernatant. The percentage of specific lysis was calculated using the formula: [(Test release − Spontaneous release)/(Maximum release − Spontaneous release)) × 100).

### 2.11. Statistical Analysis

Experimental results were reported as a mean of at least three independent experiments conducted in quintuplicates for cell viability assays and calcein-AM-release assay and triplicates for flow cytometric analyses. Data in bar graphs are presented as means ± S.E. Student’s *t*-test, one-way ANOVA, and two-way ANOVA were used for data comparison, with *p* < 0.05 as statistically significant.

## 3. Results

### 3.1. Radiotherapy Sensitizes NPC Cells to Killing by NK Cells

Induction of apoptosis in tumor cells belongs to the major direct effects of radiation on cancer. Recent studies have shown that tumor radiotherapy alters tumor immunogenicity and its interaction with the host, including an increase of NK cell-induced killing of different tumor types [23]. To estimate the influence of radiation on the killing of NPC cells by NK cells, tumor cells were irradiated with doses from 0 to 6 Gy and afterward cultured for 24, 48, and 72 h, respectively. Cells were then labelled with calcein, and subsequently co-incubated with NK cells. After 4 h of co-incubation, the concentration of calcein in the supernatant was measured as a marker for NPC cytotoxicity. Co-incubation of NPC cells with NK cells significantly induced killing in all NPC cell lines ranging from 32.6 ± 4.8% for cell line HK1 to 45.2 ± 6.3% for C666-1. When the cells were irradiated 24 h before co-incubation, NK cell cytotoxicity against NPC cells increased additionally up to 27% compared to untreated cells. The dosage of 2 Gy, commonly used as a single dose in fractionated radiotherapy, had been identified to significantly increase tumor cell killing to 46.2 ± 4.8% for cell line HK1 and 62.7 ± 4.8% for C666-1 after 24 h (Figure 1A) and was chosen for the next experiments.

As we had previously demonstrated, that the activation of NK cells with IFNβ significantly increased their killing of NPC cells [15,16,24], we further investigated whether killing of NPC cells exposed to radiotherapy was augmented when NK cells were activated. In these experiments, NK cells were treated with 1000 U/mL IFNβ for 24 h and then co-incubated with tumor cells pretreated with 2 Gy. Compared to non-activated NK cells, IFNβ significantly increased NK killing against NPC cells pretreated with 2 Gy by an average of 17% in all cell lines (Figure 1B).

### 3.2. Radiotherapy Induces Upregulation of PD-L1 in NPC Cells

Radiotherapy has been shown to increase the immunogenicity of tumor cells. On the other hand, radiotherapy can also induce immunosuppressive responses limiting an effective host-tumor immune response [25,26,27,28]. To determine whether PD-L1 expression in NPC cells is modulated by radiotherapy, expression was tested 24 h, 48 h, and 72 h after radiation of NPC cells. As shown in Figure 2 radiotherapy delivered at 2 Gy led to tumor cell surface expression of PD-L1, evident 24 h after radiation, when compared with non-irradiated cells, and sustained at 48 h and 72 h (data not shown).

### 3.3. Inhibition of PD-1 Increases Killing of NPC Cells by Activated NK Cells

As we had previously demonstrated that IFNβ induces PD-1 expression on NK cells [15,16,24], we next assessed the contribution of the PD-L1/PD-1 checkpoint to the killing of NPC cells pretreated with radiotherapy by IFNβ-activated NK cells.

Irradiated NPC cells were co-cultured with IFNβ-activated NK cells pre-treated with the anti-PD-1 antibody nivolumab. Cytotoxicity was determined by the calcein release assay. As shown in Figure 3, the addition of anti-PD-1 antibody significantly increased the susceptibility of irradiated NPC cell lines and xenograft cells to the cytotoxicity of NK cells. Nivolumab did not alter the killing of non-irradiated NPC cells by IFNβ-pre-treated NK cells, nor was there an effect on the killing of irradiated NPC cells by non-activated NK cells.

### 3.4. Radiotherapy Induces PD-1 Expression in NK Cells

Radiation can upregulate PD-1/PD-L1 expression on tumor cells and immune cells, thus increasing sensitivity to therapy with PD-1/PD-L1 inhibitors [29]. To assess the influence of radiotherapy on the expression of the members of the PD-1/PD-L1 checkpoint in NK cells, we irradiated NK cells with 2 Gy and then determined expression by flow cytometry. Radiotherapy induced surface expression of PD-1 but not PD-L1 or PD-L2 24 h after radiation (Figure 4).

Since ionizing radiation-induced expression of PD-1 on NK cells as well as PD-L1 expression in NPC cells, we next investigated whether a blockade of the PD-L1/PD-1 checkpoint would increase the cytotoxicity of NK cells against NPC cells when both cells were irradiated. As shown in Figure 5, radiation of NK cells did not alter their cytotoxicity towards NPC cell lines and xenograft cells, but it enhanced their cytotoxicity towards irradiated NPC cells and xenograft cells. When the PD-1 blocking antibody nivolumab was added to irradiated NK cells, the cytotoxicity towards irradiated NPC cells was further significantly increased. Preincubation of NK cells with IFNβ before radiation significantly increased their killing against NPC cells (Appendix A). Taken together, our results show that disrupting the PD-L1/PD-1 interaction enhances the susceptibility of NPC cells to NK cell cytotoxicity in the context of radiotherapy.

### 3.5. PD-L1 Expression in NPC Cells and PD-1 Expression in NK Cells by Radiotherapy Is Induced through the NF-κB Pathway

Radiotherapy has been shown to modulate gene expression through various signaling pathways [30], such as activation of NF-κB signaling, which may upregulate PD-L1 expression on tumor cells and PD-1 expression on immune cells. We, therefore, asked the question of whether the NF-κB pathway was involved in the up-regulation of PD-L1 in NPC cells and PD-1 in NK cells after radiotherapy. To study this, NPC cells and NK cells were pre-treated with or without the NF-κB inhibitor BMS-345541 and 1 h later exposed to radiation. Activation of NF-κB was seen in NK- and NPC cells 24 h after radiation, when not treated with BMS-345541, as shown by flow cytometry (Figure 6 and Figure 7). NF-κB expression was associated with expression of PD-1 in NK cells and PD-L1 in NPC cells (Figure 6, Figure 7 and Appendix A). Incubation of cells with BMS-345541 completely suppressed the activation of NF-κB and expression of PD-L1 and PD-1. To further investigate the role of the NF-κB pathway, irradiated NPC cells were incubated with BMS-345541 and NK cell cytotoxicity against NPC cells was measured via calcein release assay. In all cell lines, the inhibition of PD-L1 via blockade of NF-κB increased killing of irradiated NK cells towards NPC cells. This effect was similar to the one of the group in which irradiated NK cells were incubated with anti-PD-1 antibody (Figure 8). These results suggest that radiotherapy upregulates PD-L1 and PD-1 via activation of the NF-κB pathway in a cell-specific way.

## 4. Discussion

In this study, we demonstrate that (1) radiotherapy sensitizes NPC cells to NK cell killing, (2) upregulates expression of PD-1 ligand (PD-L1) in NPC cells and PD-1 receptor (PD-1) in NK cells and (3) that blocking of the PD-L1/PD-1 checkpoint further increases the killing of NPC cells by NK cells in the context of radiotherapy. In addition, we show that IFNβ increases the cytotoxic activity of NK cells against irradiated NPC cells and that this effect can also be augmented by disrupting the PD-L1/PD-1 interaction. Our results point to a potential clinical benefit of introducing PD-L1/PD-1 checkpoint blockade into radiotherapy regimens for patients with NPC and suggest that the adoptive transfer of activated NK cells could be beneficial in this setting.

Radiotherapy is the major therapeutic modality for the treatment of nasopharyngeal carcinoma [4,5]. Apoptosis, necrosis and senescence in tumor cells by induction of DNA damage and prevention of tumor growth are the main consequences of radiation on tumor tissue and reflect the beneficial effect of radiation for cancer therapy [30,31]. Recent studies have demonstrated that irradiation can also activate the host–tumor immune response [32,33] leading to the induction of immunogenic cell death (ICD).

Cytotoxic T-cells and NK cells are the main effectors of a host anti-tumor response [34]. Whereas CD8+ T-cells destroy tumor cells through recognizing tumor-specific antigens in association with MHC class I molecules, NK cells function without the requirement for prolonged pre-activation, providing a first line of defense against tumor cells [35,36]. In NPC, a number of studies have shown a positive correlation between overall survival and tumor infiltration by CD3+, CD4+, and CD8+ T-cells [37]. The role of the tumor microenvironment in NPC, however, is rather complex, with differences in the subsets of tumor infiltrating lymphocytes and their functional status, further determining the effectiveness of an host anti-tumor response and prognosis [38,39,40]. As recently, the number of NK cells in the TME of NPC tumors has been shown to be of prognostic significance [41] and IFNβ, which is used as maintenance therapy in the treatment of children and adolescents with NPC, increases the ability of patients’ NK cells to kill NPC cells in vitro [16], we decided to investigate the killing of NPC cells by NK cells in the context of radiotherapy. We are aware that our study investigates only one aspect of the complex interaction of the TME in NPC, and that further studies such as determining the functional status of NK cells in NPC patients’ tumors and peripheral blood should follow.

Here, we have demonstrated that radiation of NPC cells increased their sensitivity to killing by NK cells. The underlying mechanism could be manifold and has not been the focus of this paper. For example, radiation is known to induce expression of NKG2DL [42], TRAIL-Rs [43,44,45], and FAS [46,47,48] on tumor cells, thereby enhancing the recognition of malignant cells by NK cells and their sensitivity to killing.

In addition, radiation of NK cells also increased their cytotoxicity against irradiated NPC cells, which could be further augmented by inhibiting the PD-1/PD-L1 checkpoint. As NK cells induce apoptosis in their targets, usually via the granzyme/perforine pathway or via death ligands TNFα, FasL and TRAIL, one can postulate that radiotherapy either increased the activity of the respective death effectors in NK cells or altered the balance between activating and inhibiting NK cell receptors in irradiated cells. A study analyzing the effect of low-dose irradiation on human NK cells showed that radiation-induced direct activation of NK cells against K562 and HL-60 targets by up-regulation of expression of TNFα and FASL [49]. In addition, both the granzyme and perforine gene have been demonstrated to contain an NF-κB binding site, indicating that radiation could induce their transcription through NF-κB [50,51,52].

One limitation with the use of radiation therapy is the up-regulation of immune checkpoint signals. Our results show that radiation upregulates the expression of PD-L1 in NPC cells and PD-1 in NK cells. The induction of PD-1 expression in NK cells by radiotherapy has not been described before. Similar to our previous observation that chemotherapeutics induce PD-1 expression in NK cells via NF-κB, we have shown here that radiotherapy, which is known to induce activation of NF-κB in various cellular systems [52], also induces PD-1 expression in NK cells via NF-κB. Up-regulation of PD-L1 expression in tumor cells by radiotherapy in vivo has been recently demonstrated in patients with non-small cell lung cancer and uterine sarcoma [53,54]. PD-L1 expression in uterine sarcoma biopsies obtained after 10 Gy radiation was significantly increased compared to pre-radiation specimens (5% vs. 52%).

PD-L1 up-regulation on tumor cells and PD-1 on immune effector cells may therefore allow tumors to escape from immune surveillance. Optimization of combined radiation and immune-based therapy can be achieved using a PD-L1 blocking antibody. Here, we have demonstrated that the anti-PD-1 antibody nivolumab, blocking the PD-L1/PD-1 checkpoint increased the killing of NPC cells by NK cells in the context of radiotherapy. Such an effect has been observed in various syngeneic mouse tumor models such as colon carcinoma, melanoma, or mammary carcinoma, which demonstrated that radiation increased infiltration of tumors with CD8+ T-cells, induced IFN-γ expression in these cells, with IFN-γ then causing PD-L1 expression on tumor cells, ultimately leading to CD8+ T-cell exhaustion. When anti-PD-L1 mAb was combined with radiotherapy, anti-tumor activity was enhanced and led to a significant decrease in tumor volume [25,26]. In humans, the combination of radiotherapy with an anti-PD-1 antibody increased overall survival and progression-free survival in patients with advanced or metastatic NSCLC [28].

Previously, we have shown that IFNβ, which is used as maintenance therapy in the treatment of children and adolescents with NPC, increases the ability of patients’ NK cells to kill NPC cells in vitro [15]. We have also demonstrated that IFNβ increases the cytotoxic activity of NK cells against NPC cells treated with chemotherapeutics [24]. A similar effect has now been found in our present study where IFNβ-activated NK cells show increased cytotoxic activity toward irradiated NPC cells, which points out a role for an adoptive transfer of activated NK cells during radiotherapy of NPC tumors. Such a transfer should be accompanied by PD-L1/PD-1 checkpoint blockade in order to maximize the cytotoxic effect on NPC cells.

## 5. Conclusions

In summary, our investigations indicate that radiation boosts NK cell killing against NPC cells, and that this killing can be augmented by blockade of the PD-L1/PD-1 checkpoint. Combination of radiotherapy with PD-L1/PD-1 checkpoint blockade could therefore increase the efficacy of radiotherapy in NPC by enhancing immunogenicity of tumor cells.

## Figures and Tables

**Figure 1 cells-10-02458-f001:**
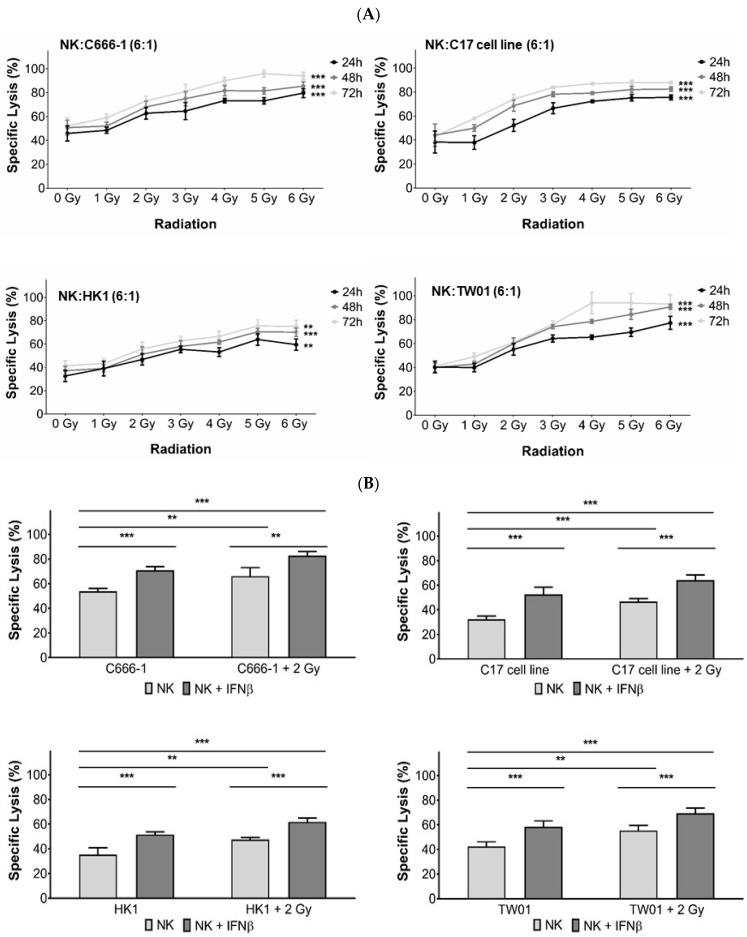
Radiotherapy sensitizes NPC cells to killing by NK cells. NPC cells were irradiated with doses from 0 to 6 Gy and cultured thereafter for 24, 48 or 72 h in medium. Cells were then labeled with calcein, plated in a 96-well plate and incubated with NK cells for 4 h at an E:T ratio of 6:1. (**A**) Lysis of target cells was determined by the measurement of calcein in collected supernatants by an ELISA reader. (**B**) NK cells were pre-incubated or not with IFNβ (1000 U/mL) for 24 h and then co-incubated with NPC cells irradiated or not with 2 Gy and cultured for 24 h as above. Data are presented as means ± S.E.M. Asterisks indicate statistically significant differences between the non-irradiated group and irradiated groups in all cell lines (a: two-way ANOVA; b: one-way ANOVA; ** *p* < 0.01; *** *p* < 0.001).

**Figure 2 cells-10-02458-f002:**
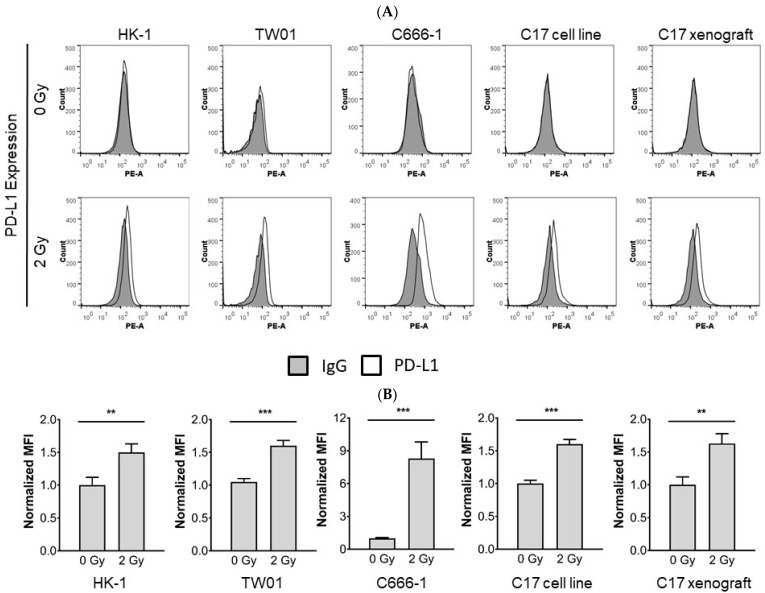
Radiotherapy induces surface expression of PD-L1 in NPC cells. PD-L1 surface expression was measured by flow cytometry 24 h after radiation of NPC cells with 2 Gy. (**A**) Histogram plots of PD-L1 expression on NPC cells; (**B**) Quantification of PD-L1 expression on NPC cells. Quantitative data of three independent experiments is shown as normalized mean fluorescence intensity (MFI). Normalized MFI is calculated by dividing the MFI of the anti-PD-L1 antibody-stained sample by the MFI of the control antibody. Data is shown as mean ± S.E.M. Asterisks indicate statistically significant differences between all cell lines (one-way ANOVA; ** *p* < 0.01; *** *p* < 0.001).

**Figure 3 cells-10-02458-f003:**
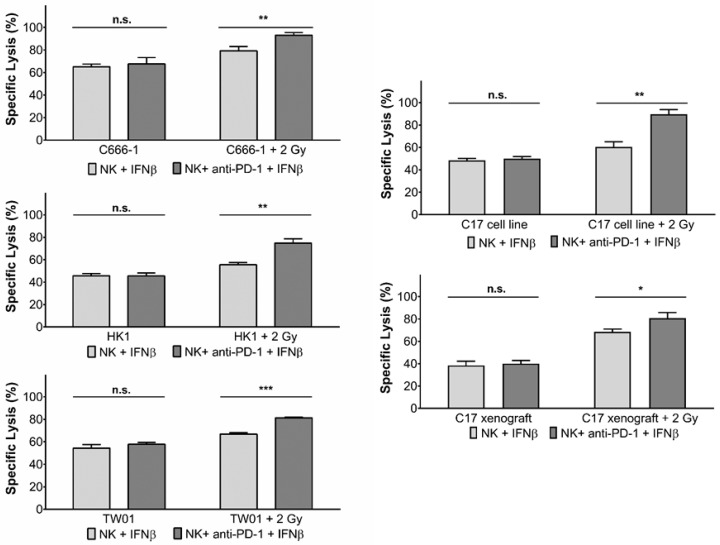
Inhibition of PD-1 increases killing of NPC cells by activated NK cells. Pre-treatment of IFNβ-activated NK cells with the anti-PD-1 antibody nivolumab for 24 h and subsequent exposure to NPC cells irradiated 24 h before with 2 Gy. E:T ratio 6:1. Cytotoxicity was measured by calcein release assay. Data are presented as means ± S.E.M. Asterisks indicate statistically significant differences between all cell lines (two-way ANOVA; * *p* < 0.05; ** *p* < 0.01; *** *p* < 0.001; n.s.: not significant).

**Figure 4 cells-10-02458-f004:**
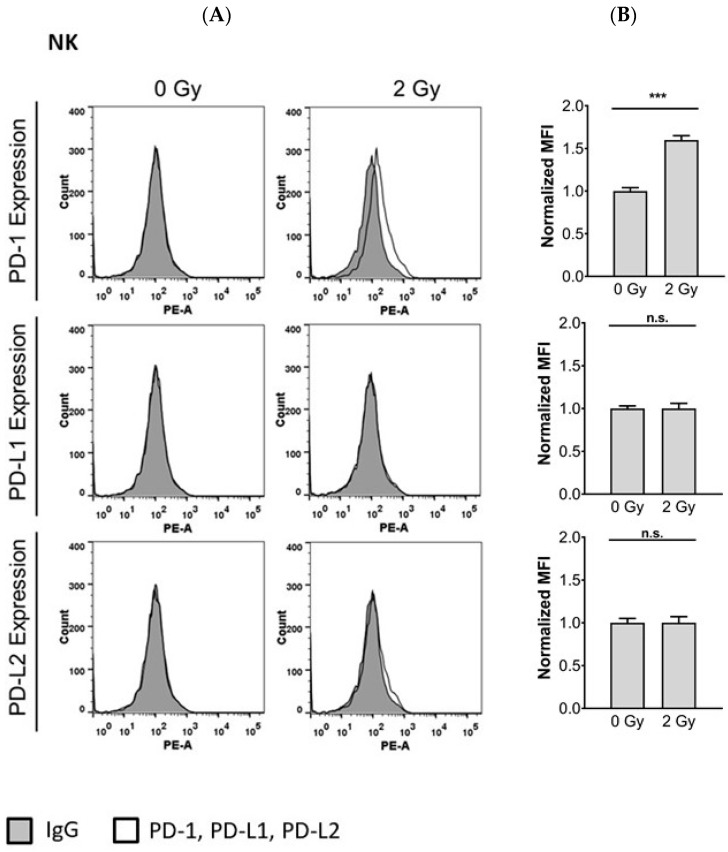
Radiotherapy upregulates surface expression of PD-1 in NK cells. Cells were irradiated with 2 Gy. After 24 h PD-1, PD-L1, and PD-L2 expression was analyzed by flow cytometry. (**A**) Histogram plots of PD-1, PD-L1 and PD-L2 expression on NK cells; (**B**) quantification of PD-1, PD-L1 and PD-L2 expression on NK cells. Quantitative data of three independent experiments are shown as normalized mean fluorescence intensity (MFI). Data of normalized MFI is calculated by dividing the MFI of the sample stained by the antibody of interest by the MFI of the control antibody. Data is shown as mean ± S.E.M. Asterisks indicate statistically significant differences between all cell lines (one-way ANOVA; *** *p* < 0.001; n.s.: not significant).

**Figure 5 cells-10-02458-f005:**
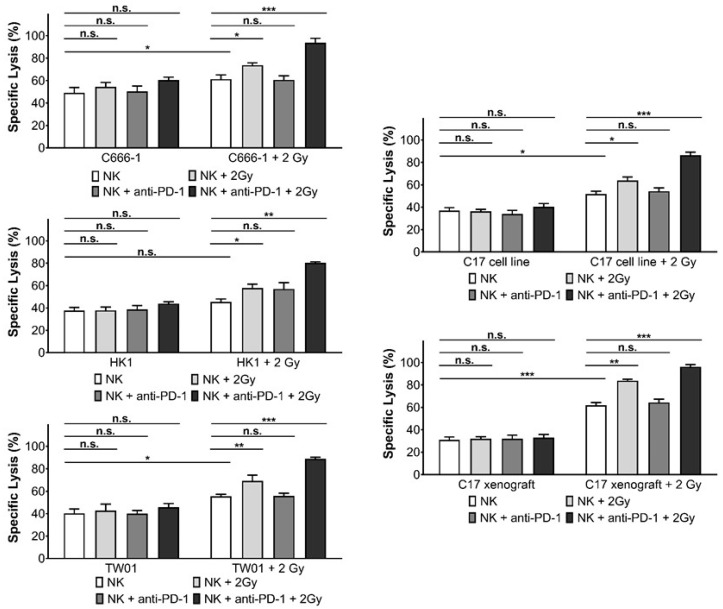
Inhibition of PD-1 increases killing of NPC cells by NK after radiation. NPC and NK cells were irradiated with 2 Gy. 24 h later, target cells were labeled with calcein, plated in a 96-well plate and incubated with NK cells for 4 h at an E:T ratio of 6:1. Lysis of target cells was determined by measurement of calcein in collected supernatants by an ELISA reader. Data are presented as means ± S.E.M. Asterisks indicate statistically significant differences between all cell lines in one ratio-group (two-way ANOVA; * *p* < 0.05; ** *p* < 0.01; *** *p* < 0.001, n.s.: not significant).

**Figure 6 cells-10-02458-f006:**
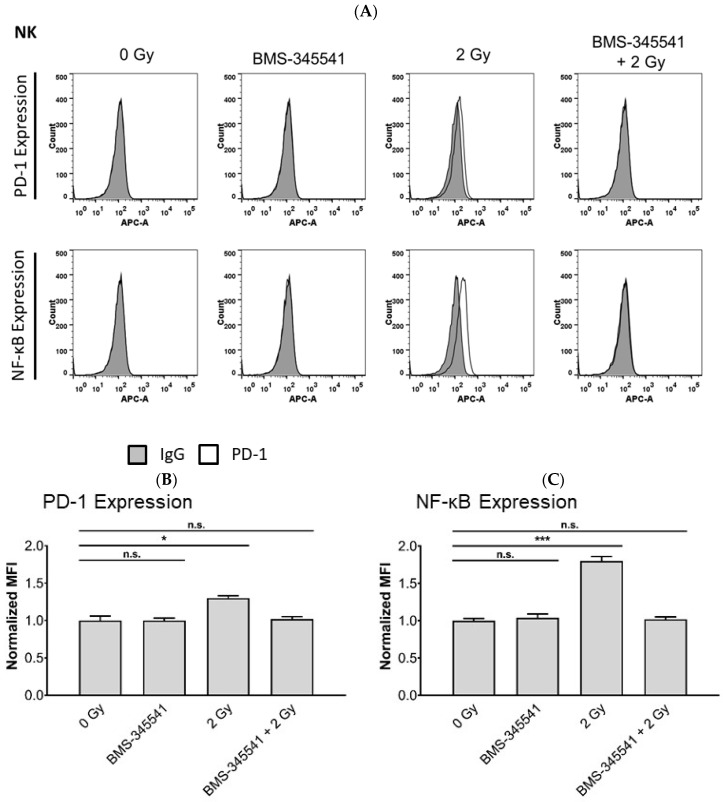
Radiotherapy induces PD-1 expression in NK cells via upregulation of NF-κB. NK cells were incubated with the NF-κB inhibitor BMS-345541 for 1 h before radiation. Expression of NF-κB, PD-L1 and PD-1 was analyzed by flow cytometry. (**A**) Histogram plots of PD-1 and NF-κB expression on NK cells. Quantification of PD-1 (**B**) NF-κB (**C**) expression on NK cells. Quantitative data of three independent experiments is shown as normalized mean fluorescence intensity (MFI). Normalized MFI is calculated by dividing the MFI of the sample stained with the antibody of interest by the MFI of the control antibody. Data is shown as mean ± S.E.M. Asterisks indicate statistically significant differences between all cell lines (two-way ANOVA; * *p* < 0.05; *** *p* < 0.001).

**Figure 7 cells-10-02458-f007:**
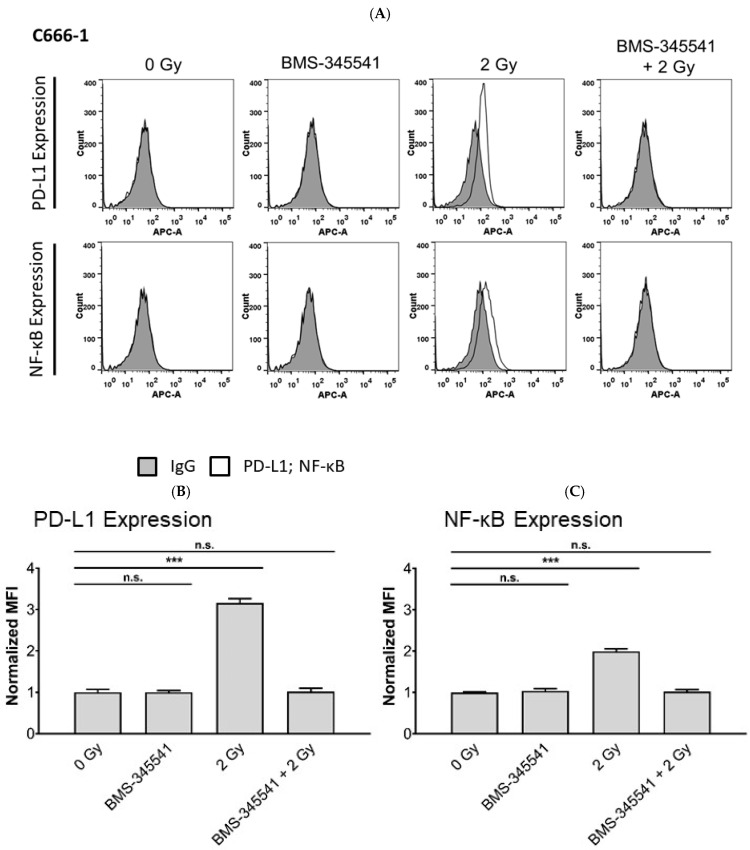
Radiotherapy induces PD-L1 expression in NPC cells via upregulation of NF-κB. C666-1 cells were incubated with the NF-κB inhibitor BMS-345541 for 1 h before radiation. Expression of NF-κB, PD-L1 was analyzed by flow cytometry. (**A**) Histogram plots of PD-L1 and NF-κB expression on NPC cells; quantification of PD-1 (**B**) and NF-κB (**C**) expression on NPC cells. Quantitative data of three independent experiments are shown as normalized mean fluorescence intensity (MFI). Normalized MFI is calculated by dividing the MFI of the stained sample by the MFI of the control antibody. Data is shown as mean ± S.E.M. Asterisks indicate statistically significant differences between all cell lines (two-way ANOVA; *** *p* < 0.001; n.s.: not significant).

**Figure 8 cells-10-02458-f008:**
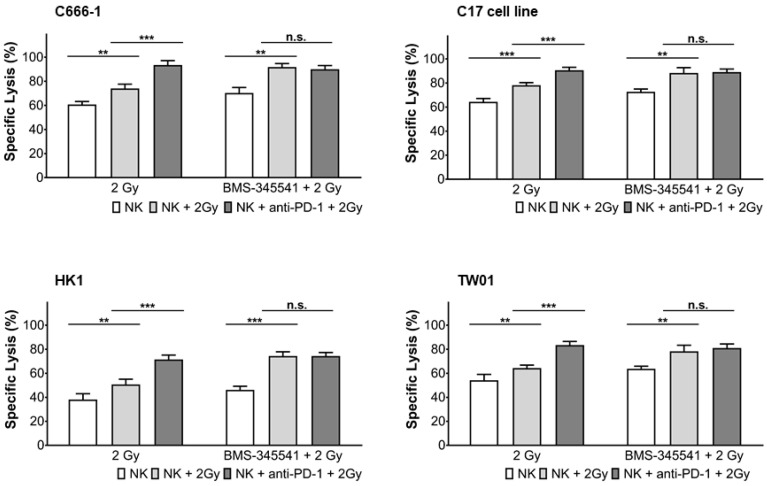
Inhibition of NF-κB increases killing of NPC cells by NK cells after radiation. NPC and NK cells were irradiated with 2 Gy. NK cells were treated or not with the NF-κB inhibitor BMS-345541 for 1 h before radiation. 24 h later, target cells were labeled with calcein, plated in a 96-well plate and incubated with NK cells for 4 h at an E:T ratio of 6:1. Lysis of target cells was determined by measurement of calcein in collected supernatants by an ELISA reader. Data are presented as means ± S.E.M. Asterisks indicate statistically significant differences between all cell lines (two-way ANOVA; ** *p* < 0.01; *** *p* < 0.001; n.s.: not significant).

## Data Availability

The raw data supporting the conclusions of this article will be made available by the authors, without undue reservation.

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
