# Peer review of "Radiotherapy Combined with PD-1 Inhibition Increases NK Cell Cytotoxicity towards Nasopharyngeal Carcinoma Cells"

_cells, 2021, doi:10.3390/cells10092458_

Round 1
Reviewer 1 Report
The authors describe a novel finding that radiotherapy enhances NK cell cytotoxic activity against NPC cells, and that this can be augmented further by the use of PD-L1/PD-1 checkpoint inhibitors, suggesting an interesting idea that combinatorial radiotherapy and immune checkpoint inhibitor treatment could enhance tumour immunogenicity.
This is an exciting and timely article that will make a significant contribution to the field and has transformative potential for current NPC treatment regimes.
There is one significant point requiring clarification:
- Line 143 - where did you get the ratio from? Was this previously tested by you? Previously published? Or optimised as part of your experimental procedures? Please confirm and ensure this is clarified in the text.
There are a few minor typos and issues to address as follows:
- Abstract, line 18 - start a new sentence after "assay" (or at the very least, separate the two clauses with a semi-colon
- Introduction, line 36 - please rephrase to "...has allowed the reduction of..." for clarity's sake
- Materials & Methods, line 64 - I believe it's Prof. Kwok Wai Lo (not Lo Kwok Wai) - please double check and alter accordingly
- Line 75 - please change to "cell" (singular)
- Line 116 - please remove the space before the colon after "DMSO"
- Line 128 - when starting a sentence with a number, please write out in full (in this case "Two")
- Results - all subheadings should be numbered appropriately (e.g., 3.1, 3.2, etc.)
- Lines 196-197 - "On the other side" is an odd choice of phrase - please rephrase to "On the other hand"
- Line 200 - please change "lead" to "led" (past tense)
- Figures 2, 4, 6 & 7 - the figure legends are rather brief. Please expand as per the journal guidelines to include a brief explanation (like you have for other figures).
- Line 324 - please change "lead" to "led" (past tense)
- Line 339 - please remove "to" so it reads "...points out a role for..."
Once these have been addressed, I see no reason why this shouldn't be published in Cells.
Reviewer 2 Report
In the manuscript by Makowska et. al., the authors investigate the impact of radiotherapy on NK-cell mediated killing of NPC cells. Irradiation of both NK and NPC cells leads to increased expression of immune checkpoint inhibitors. Further, targeting the PD-L1/PD-1 pathway leads to enhanced killing of cancer cells by NK cells in an Nf-κβ-dependent manner. The results have been presented clearly using an in vitro system, however, the significance of the entire findings is not very promising in absence of animal or clinical data. Nevertheless, the findings of the manuscript are very interesting, and supplementing the data with an in vivo or ex vivo assay using an animal tumor model will increase the translational value of the work. The authors should address the following concerns before publication:
- The manuscript focuses on NK-cell mediated killing of NPC cells. While the NK cells are a critical component of altered tumor microenvironment upon radiotherapy treatment, the other major well-characterized immune mechanisms like cancer cell killing mediated by CD8T cells is not discussed. While it is difficult to establish the significance of NK cells in absence of in vivo data or patient data, the authors should at least discuss all these aspects in the discussion part of the manuscript.
- The radiotherapy treatment in NPC cells leads to both enhanced killing capacity of NK cells and increased expression of checkpoint inhibitors which is intriguing. While the authors further show that blocking PD-L1 pathways potentiates the killing capacity of NK cells, it is not clear how radiotherapy increases the killing capacity of NK cells? Can the authors comment on this?
- There are multiple expression analyses performed by FACS that are not quantified. The authors should quantitate the data shown in Fig. 2,4, 6, and 7.
- The authors have used two-way ANOVA to compare statistical differences for most of the data. However, in methods, they have described using Student’s T-test only. The details of all statistical tests should be included in the methods. The details of antibodies should also be provided in the methods.
- In Fig 1B, the authors should use one-way ANOVA and compare all 4 groups with each other.
- In Fig 5, were the untreated NPC cells significantly different in comparison to 2Gy treated NPC cells especially, C666-1, HK1, and TW01 cells.
- In Fig 5, the authors show that irradiated NK cells when treated with anti-PDL1 antibody, increases the killing capacity of NK cells. Does treatment of irradiated NK cells also lead to an increase in killing activity when treated with IFNβ?
- In Fig 8, can the authors include control NPC cells co-cultured with NK cells as a positive control? Can the authors perform an in vivo experiment in mice to validate their in vitro findings?
Round 2
Reviewer 2 Report
The authors have addressed most of the raised concerns satisfactorily. While for some of the comments, it is understandable that authors don't want to extend the scope of study to address significance and would like to follow that in future. The revised manuscript should be accepted.